# Synthesis and Investigation of the Properties of Biphasic Hybrid Composites Based on Bentonite, Copper Hexacyanoferrate, Acrylamide and Acrylic Acid Hydrogel

**DOI:** 10.3390/polym15122586

**Published:** 2023-06-06

**Authors:** Galymzhan Kulamkadyrovich Mamytbekov, Dmitry Anatol’evich Zheltov, Yernat Rashidovich Nurtazin

**Affiliations:** Institute of Nuclear Physics, Ibragimov Street 1, Almaty 050032, Kazakhstan; d.zheltov@inp.kz (D.A.Z.); nurtazin@inp.kz (Y.R.N.)

**Keywords:** biphasic hybrid composite materials, liquid radioactive waste (LRW), polymer and mineral matrix, copper hexaferrocyanide, sorption

## Abstract

This article presents a study of the synthesis and characterization of new biphasic hybrid composite materials consisting of intercalated complexes (ICC) of natural mineral bentonite with copper hexaferrocyanide (phase I), which are incorporated into the bulk of the polymer matrix (phase II). It has been established that the sequential modification of bentonite with copper hexaferrocyanide and introduction of acrylamide and acrylic acid cross-linked copolymers into its volume by means of in situ polymerization promote the formation of a heterogeneous porous structure in the resulting hybrid material. The sorption abilities of prepared hybrid composite toward radionuclides of liquid radioactive waste (LRW) have been studied, and the mechanism for binding radionuclide metal ions with the components of the hybrid composition have been described.

## 1. Introduction

One of the important areas in the development of the modern chemistry of synthetic and natural polymers is the creation of composite materials based on natural micro- and nano-fillers, providing the products with improved physical-mechanical and barrier properties, high sorption activity and selectivity toward certain metal ions and other substrates as well. Among all potential nanocomposite precursors, those based on clay and layered silicates have been more widely investigated, since the starting clay materials are easily available and their intercalation chemistry has been studied for a long time [1,2]. Due to micro- and nanometer-size particles resulting from dispersion, these nanocomposites can have substantially improved mechanical, thermal, optical and physical-chemical properties in comparison with the pure polymer or conventional (microscale) composites.

Low-level radioactive waste (LRW) is generated as a result of nuclear reactor operations and maintenance, mining and milling of nuclear fuel and spent nuclear fuel recycling. In general, such radioactive waste contains a variety of radionuclides that are fission byproducts resulting from the above processes. There is a need for materials and processes capable of recovering radionuclides from low-metal-concentration solutions containing, in some cases, high salt levels (for example, sea water or alkaline solutions) [3]. Waste storage tanks currently provide a stable repository for LRW, but high salt content can in the long term corrode the tank structural materials, and this poses serious potential environmental risks near these facilities.

To prevent the spread of easily migrating forms of radionuclides in LRW (^134^Cs^+^, ^137^Cs^+^, ^85, 90^Sr^+2^, ^90^Y^+3^, ^57, 60^Co^+2^), it is recommended to use materials with acceptable values of thermal and radiation stability, mechanical strength, sorption activity, and neutrality to the action of acids, alkalis and solvents. An important aspect is the controllability and predictability of the process of radionuclide sorption in the volume of a natural or synthetic matrix. The matrix materials should have acceptable physical parameters for application as structural backfill materials and the creation of artificial barriers in the construction of engineering structures [4,5].

As a rule, the choice of the most appropriate method for LRW management is determined by technical and non-technical factors. It should be noted that there are practically no materials that satisfy all requirements of radiation safety both in the environment and in the places of their direct production—at nuclear power plants (NPP). A rationally justified way to meet these requirements is the development of a technology for the synthesis of new classes of composite materials consisting of natural and synthetic polymeric materials [6,7,8,9].

It is well known that clays, some classes of polymers, and hexaferrocyanide complexes of transition metals are currently the most studied classes of natural and synthetic sorbents for the immobilization of radionuclides from LRW [10,11,12,13,14,15,16,17].

This paper covers: (a) the synthesis and characterization of a new biphasic hybrid composite material consisting of intercalated complexes (ICC) of the natural mineral bentonite with transition metal hexaferrocyanides (phase I), which are incorporated into the bulk of the polymer matrix (phase II); (b) their sorption properties toward radionuclides (RN) of liquid radioactive waste.

To date, there are no reports about direct encapsulation of radioactive waste into biphasic hybrid compositions in which the polymer network is the main carrier matrix in a volume in which intercalated complexes of clay-copper hexaferrocyanide are distributed.

The aim of the study was to analyze immobilization of LRW radionuclides from the research water-water reactor WWR-K at the Institute of Nuclear Physics (Almaty) by a hybrid composite consisting of an intercalated bentonite–copper hexaferrocyanide complex in the body of a cross-linked (acrylamide-acrylic acid) copolymer matrix.

## 2. Experimental Section

### 2.1. Materials

Monomers of acrylic acid (AAc), acrylamide (AAm) and crosslinking agent N,N′-methylene-bis-acrylamide (MBAAm) and initiators ammonium persulfate (NH_4_)_2_S_2_O_8_ (PSA) and sodium metabisulfite Na_2_S_2_O_3_ (SMB) from Sigma-Aldrich (USA) of chemically pure grade were used with no additional purification. Bentonite of the JSC “Ilsky Zavod Utyazhelitel” (Russia) TU-2164-003-00136716-2015 was used after washing with distilled water and drying to a constant weight at 70 °C temperature.

Copper vitriol CuSO_4_·5H_2_O and potassium hexaferrocyanide K_4_[Fe(CN)_6_]·3H_2_O of chemically pure grade were used with no further purification.

### 2.2. Preparation of Composite Materials

ICC was synthesized by mixing aqueous solutions of CuSO_4_ (0.1 M) and K_4_[Fe(CN)_6_] (0.1 M) in a ratio of 2:1 in a volume of bentonite swollen in water with vigorous stirring on a magnetic stirrer. After washing and filtering the suspension, the solid residue of ICC was dried at 60 °C to constant weight.

Synthesis of PCC consisted of the following stages: 5 g of ICC sample swollen in water, 5 g of AAm, 5 mL of AAc and the calculated amount of MBA (5 mL of 0.1 M, which corresponds to the degree of crosslinking of the polymer gel of 0.72 mol.% or 0.01 wt.% in relation to the acrylamide monomer) were added under stirring. The total volume of the reaction system was 100 mL. At the final stage of stirring, 0.5 mL of APS and 0.5 mL of SMB were added to the reaction mixture at a concentration of 0.1 M, respectively. After thorough mixing, the reaction mixture was poured onto rectangular and cylindrical substrates made of organic material and placed in an oven at 60 °C temperature, where in situ polymerization proceeded for 6 h, after which the samples were cooled to room temperature during the day. After the samples were removed from the substrates, they were repeatedly washed and dried in a thermostat to a constant weight at t < 50 °C to prevent thermal degradation of the polymer matrix.

### 2.3. Instruments and Measurements

The experiments on sorption of LRW radionuclides by a natural mineral and its percolated analogue were performed under static conditions by periodic stirring of the sorbent samples placed in 21 mL of LRW for 24 and 72 h. Then, the liquid and solid phases were separated in a separating funnel to prevent the sorption of RN on a paper filter. In the resulting filtrate, the specific activity of RN was determined, and the degree of binding (purification) *θ* and the distribution coefficient *K_d_* of radionuclides between the liquid and solid phases were calculated according to Equations (1) and (2), respectively [18,19].
(1)$$\theta =\frac{A_{0}-A_{eq}}{A_{0}}\cdot 100\%$$
(2)$$K_{d}=\frac{A_{0}-A_{eq}}{A_{eq}}\cdot \frac{V}{m_{S}}(\frac{mL}{g})$$
where *A*_0_ and *A_eq_* are the initial and equilibrium specific activity of RN before and after sorption, respectively; *V* is the volume of the liquid phase, mL; *m_s_* is the sorbent mass, g.

The activity of the radionuclides was measured on a gamma-spectrometer using a BE3820S coaxial germanium detector by Canberra (USA). The measured energy resolution of the detector was 1.7 keV at the half-height peak at an energy of 1332 keV (60Co).

To study swelling, a hybrid hydrogel composite with a size of (0.5 × 0.5) cm^2^ was immersed in distilled water at regular time intervals, kept and weighed. Before weighing, water was removed from the surface using tissue paper. The swelling coefficient (*K_sw_*) was calculated using the following formula:(3)$$K_{sw}=\frac{m_{w}-m_{0}}{m_{0}},g\cdot g^{-1}$$
where *m_w_* and *m*_0_ are the weights of wet and dry hydrogel composite, respectively.

For elemental analysis of leachates and water samples (if necessary, diluted with deionized water) the following methods were used:-inductively coupled plasma mass spectrometry (ICP MS) on the ELAN-9000 quadrupole mass spectrometer (PerkinElmer SCIEX, Waltham, MA, USA);-inductively coupled plasma optical-emission spectrometry (OES-ICP) on the OPTIMA-8000 double-view OE spectrometer (PerkinElmer Inc., Waltham, MA, USA).

Scanning electron microscopy (SEM) was performed using a Hitachi model TM 4000 Plus microscope equipped with an X-ray fluorescence energy dispersive analysis (EDS) attachment with a crystal detector. The samples were studied under low vacuum in the back-scattered electron mode.

X-ray diffraction analysis (XRD) was performed on a Bruker X-ray diffractometer with CuK_α_ radiation. A thin dried film of a hybrid composition and powdered bentonite up to 10 µm in size were scanned in the 2θ angle range from 20° to 70° with a step of 0.02°, a scanning speed of 2 s/point, and radiation parameters of 40 kV and 40 mA.

For the MS investigations, the samples were mixed with paraffin at a rate of 100 mg/cm^2^ and transferred into tablets. The Mössbauer spectra of the samples were recorded in the transmission geometry. The source of γ-quanta was ^57^Co in the chromium matrix. The spectra were recorded at 300 K on a nuclear gamma-resonance spectrometer MS1104Em. The reference sample α-Fe was used for spectra calibration. The Mössbauer spectra were processed using the software SpectrRelax by the methods of model fitting [19].

## 3. Result and Discussions

### 3.1. Characterization of Composite Polymeric Material

#### 3.1.1. FT-IR Spectroscopy

Figure 1 shows the FT-IR spectra of BT (1), its intercalated complex (ICC) with copper hexaferrocyanide BT-K_(4−x)_Cu_x_[Fe(CN)_6_], percolated in the volume of the cross-linked copolymer of acrylamide and acrylic acid (P[AAm-AAc]) ICC (3) and potassium hexaferrocyanide (PHF) (4). The presence of AAc and AAm units in the polymer composites is confirmed by intensive peaks in the absorption region of 1600–1750 cm^−1^, corresponding to the valence vibrations of the carbonyl group -C=O and the amide bond -CO-NH_2_. Fluctuations in the frequency region of 1560 cm^−1^ and 1450 cm^−1^ indicate the presence of carboxylate groups -COO- in the polymer matrix [20,21].

All compounds are characterized by a broad absorption band in the range of 3000–3626 cm^−1^ related to the valence vibrations of OH, -NH_2_ groups of the polymer matrix and free water as well as the systems of hydrogen bonds formed between functional groups of the polymer matrix (-COOH⋯CONH_2_) and oxygen groups of the mineral matrix and polymer (-COOH⋯O-Si-O (OAl-OH). The peaks in the absorption region of 1600–1640 cm^−1^ can be attributed to the deformation vibration of the OH groups of bentonite and polyacrylamide matrices, respectively, and to the adsorbed water as well.

The presence of hexaferrocyanide complexes of copper in the modified aluminosilicates is evidenced by the complex absorption bands in the region of 2000–2150 cm^−1^. The absorption peak at 2087–2092 cm^−1^ corresponds to the valence vibrations of nitrile groups (-C≡N) and most likely refers to the mixed hexaferrocyanide phase both of potassium and copper. The band at 2042 cm^−1^ corresponds to the vibrational spectra of K_4_[Fe(CN)_6_]·3H_2_O.

In the IR Fourier spectra of the studied samples, absorption bands were also recorded, which are characteristic of the C≡N valence vibrations (2340 cm^−1^ and 2360 cm^−1^), the intensity of which increased especially for intercalated and percolated hexaferrocyanides in the mineral (curve 2) and mineral-polymer (curve 3) phases compared to pure potassium hexaferrocyanide (curve 4) [22,23].

Despite the wide variety of spectra for silicates of various structures and compositions, all of them have sharp absorption bands in the region of 800–1300 cm^−1^, which explains the spectra in the region of these frequencies based on the properties of silica and alumino-oxygen groups only.

In aluminosilicate systems, the presence of very weak bands at 916 cm^−1^ can be explained by the appearance of deformation vibrations of the Al-O(H) non-bridging bond, which is longer and weaker than the Si-O bond [22]. The interatomic distances of Si-O and O-O in the same tetrahedron in silicate structures are not the same. The high strength of Si-O groups does not make them rigid; on the contrary, they are highly adaptable to the habit-morphological conditions created by other structural elements, leading to their significant deformation.

When cations with a large ionic radius enter the structure of silicates, the maxima of the absorption band are shifted to the long-wave region. Layered silicates have one intense band at 1012 cm^−1^ and weaker bands around 1116 cm^−1^ and 909 cm^−1^. Substitution of aluminum for magnesium or iron leads to a bathochromic shift of the strong Si-O band. It should be noted that percolation of polymer chains into the volume of mineral matrix aggravates the process of rearrangement of the internal structure of bentonite and its intercalated complex with copper ferrocyanide. It can be seen that introduction of polymer chains into the mineral matrix by in situ polymerization leads to a shift of the main absorption band of the valence vibrations of Si-O-groups to a higher wavelength region, in particular, from 1012 cm^−1^ to 1037 cm^−1^.

The band at 798 cm^−1^ corresponds to the stretching symmetric vibrations of Si-O-Si and Si-O-Al, and 1010–1030 cm^−1^ and 1052 cm^−1^ can be attributed to the Si-O- groups of the silicon-oxygen tetrahedrons. The band at 3690 cm^−1^ is associated with vibrations of the Al-OH-O grouping, where O is oxygen from the neighboring layer, while the Al-OH-O grouping, where O is oxygen from the inner layer bound to the [SiO_4_]^4−^ tetrahedron gives a band at 3623 cm^−1^. Since the Si-O bonds are the strongest in the structure with a relatively high percentage of covalent character, their individuality can be recognized in the infrared spectra of such complex structures even if the corresponding absorption bands are weakly influenced by the ions in the second coordination sphere [23].

The IR spectra of aluminosilicates are very sensitive to substitution of ions with different charge values; for example, A1^3+^ instead of Si^4+^ at coordination number Al = 4 or Mg^2+^ instead of A1^3+^, or vice versa, at coordination number A1 = 6, are less sensitive to substitution of ions with the same charges [24,25].

Introduction of trivalent ions into the sites with tetrahedral coordination, for example, A1^3+^ instead of Si^4+^, in all cases causes a shift in the absorption bands of the main valence vibration of Si-O (region from 900 to 1100 cm^−1^) toward slightly lower frequencies due to an increase in the average (Si, Al)-O distance. The frequency of the stretching vibration of Si-O bond decreases linearly with increasing proportion of aluminum ions in the position with tetrahedral coordination [26].

#### 3.1.2. Mössbauer Spectroscopy

Figure 2 shows the Mössbauer spectra (MS) of potassium hexaferrocyanide (1), bentonite (2), the complexes of potassium hexaferrocyanide (3) and copper hexaferrocyanide intercalated in bentonite (4), as well as the percolated complex P[AAm-AAc]:{BT-K_(4−x)_Cu_x_[Fe(CN)_6_]} (5). The MS spectrum of potassium ferricyanide is a non-broadened singlet peak with a chemical shift of δ = −0.045 mm/s, which noticeably simplifies the analysis of gamma-spectra of the intercalated mineral and its percolated form in polymer matrix composition [26,27]. The absence of quadrupole splitting in the gamma resonance spectra of potassium cyanoferrate indicates a cubic structure or close to its symmetry. The electronic structure of the Fe^2+^ ion can be represented as 3d^6^4s^0^4p^0^, with the sixth d-electron making the main contribution to the magnitude of the electric field gradient.

In the most common octahedral environment of Fe^2+^ with negatively charged ions, the ${3d}_{z}^{2}$ and $3d_{x^{2}-y^{2}}$ orbitals of iron extended along the coordinate axes toward the cyanide (CN^−^) ligands are energetically less advantageous than the 3d_xy_, 3d_xz_, and 3d_yz_ orbitals located between the coordinate axes. These three orbitals are degenerated and statistically equally occupied by the sixth d-electron. Such a uniform distribution has cubic symmetry [26,27,28,29] and does not create an electric field gradient.

Thus, the result is a spherically symmetric charge distribution, the electric field gradient in all directions is the same, and there is no quadrupole splitting. It is characteristic that no quadrupole splitting of the ferricyanide phase (D4) is observed for both intercalated and percolated complexes containing copper ferricyanide (K_(4−x)_Cu_x_{Fe(CN)_6_]. It was noted earlier that in transition metal hexacyanoferrates, the environment of the central iron ion is similar to that of the cyanoferrate ion. The external sphere cation is surrounded by *x* nitrogen atoms of the CN^−^ group and six water molecules [29,30,31].

It is known [32] that the structure of layered bentonite silicate consists of densely packed large oxygen and hydroxyl anions and small-size cations in tetrahedral (T) or octahedral (O) coordination forming, respectively, tetrahedral or octahedral networks. The bentonite is a three-layered package constructed of two tetrahedral layers with one octahedral layer or 2:1 type layer between them.

The isomorphism of cations in tetrahedral coordination Si^4+^ to Al^3+^ and *R*^3+^ to *R*^2+^ (for dioctahedral layers) or 2*R*^2+^ to *R*^3+^ (for trioctahedral layers) in octahedral positions (*R*^n+^ is an octahedral cation of certain valence) is characteristic of 2:1 structure. Such substitutions are non-stoichiometric and create a deficit in positive charges, which are compensated from interlayer positions predominantly by monovalent, and less often by divalent, large cations. The general composition of layers with a 2:1 structure, which differ in layer charges, can be represented by the formula [33,34]:(4)$$R_{\frac{x+y+z}{n}}^{n+}=\{{{\left(R^{3+}R^{2+}R^{+}\right)}_{2-3}^{6-x}{[{\mathrm{Si}}_{4-y}{\mathrm{Al}}_{y}}_{4}O_{10}]({\mathrm{OH}}_{2-z})\}}^{x+y+z}$$
where *x* + e + *z* is the charge of the layer, which is usually determined from the recalculation of chemical analyses or from capacity values, although it is not possible to obtain an objective estimate in this case [34].

Almost all oxygen atoms of adjacent networks contact each other at the minimum potential energy of the intermolecular potential field formed by the total dipole effect of the layer itself. The rings of the oxygen atoms of the tetrahedron bases are close to or slightly different from the hexagonal symmetry, which is generally characteristic of most trioctahedral layers [35,36,37].

The Mössbauer spectra of iron-containing bentonite are a superposition of three quadrupole doublets (Figure 2). The spectrum of Fe^2+^ is decomposed into a doublet at the M1 and M2 positions, the first of which corresponds to octahedrons with OH^−^ groups in the trans-position, whereas the second one characterizes the location of hydroxyl groups in the cis-position of octahedrons, with twice as many octahedrons in M2 position than in M1 [32,33]. The well-resolved doublets appear with chemical shifts at δ = 1.225 and 1.137 mm/s and quadrupole splitting Δ = 1.785 and 2.750 mm/s, respectively. The quadrupole doublet with greater splitting (2.750 mm/s) can be assigned to Fe^2+^ in small M2 octahedrons, while the doublet with lesser quadrupole splitting (1.785 mm/s) corresponds to Fe^2+^ ions in large, usually unoccupied M1 octahedrons.

According to the literature data [35,36,37], the Fe^2+^ ions in aluminosilicates mainly occupy the octahedral positions M1 and M2, which are characterized by the trans- and cis- configurations of the OH group pair positions, respectively. Occupancy of M2 positions by these ions leads to large lattice distortions and, therefore, these positions are assigned the most intensive doublet with large quadrupole splitting in the Mössbauer spectra. Thus the doublet with smaller contribution of the chemical shift and Δ is attributed to Fe^2+^ ions in the M1 positions. The doublets corresponding to Fe^3+^ ions are also assigned to different structural positions: with a large Δ- to octahedral M2, and with a smaller Δ- to octahedral M1 or tetrahedral [32].

The Fe^2+^/Fe^3+^ ratio in the studied samples decreased from 1.94 in pure bentonite to 1.43 for the hybrid composition. Moreover, the content of Fe^3+^ slightly changed during the bentonite clay modification, while the content of Fe^2+^ shows a noticeable decrease. The content of [Fe(CN)_6_]^2−^ ion was also evident for transition from the BT-K_4_[Fe(CN)_6_] system to BT-K_(4−x)_Cu_x_[Fe(CN)_6_] and the percolated hybrid complex P[AAm-AAc]:{BT-K_(4−x)_Cu_x_[Fe(CN)_6_]}, with a simultaneous displacement in chemical shift toward more negative values, but the quadrupole splitting remained zero. This may have been due to the coexistence in these compounds of individual molecules of K_4_Fe(CN)_6_] and the hexaferrocyanide copper complex as well. This conclusion was previously reached by the authors [38].

The value of the isomeric shift allowed us to estimate the coordination number of iron atoms. Chemical shifts for Fe^2+^ and Fe^3+^ showed a noticeable tendency to decrease with decreasing coordination number. The inner doublet for Fe^2+^ with lower quadrupole splitting (1.785 mm/s) had an intensity approximately twice lower than the outer one and was attributed to M1 octahedrons, the number of which in the structure was two times lower than M2. The latter corresponded to the doublet with a large value Δ (2.750 mm/s). The spectrum contained one doublet of trivalent iron Fe^3+^ with δ = 0.351 mm/s and quadrupole splitting Δ = 0.538 mm/s in the M2 octahedron.

The Fe^3+^ ions showed a decrease in isomeric shift from 0.351 mm/s in pure bentonite to 0.251 mm/s when potassium hexaferrocyanide (0.311 mm/s) or copper hexaferrocyanide (0.251 mm/s) was incorporated into the mineral matrix. This may have been associated with an increase in covalent bonding in the tetrahedral complexes with an increase in the 4s-electron density in the region of the Fe^2+^ iron nucleus. In addition, as shown in Table 1 and Figure 2, in the percolated form of the composition along with Fe^3+^, a new band appeared at δ = 0.395 mm/s with a quadrupole splitting of Δ = −0.186 mm/s, which could be attributed to the FeO_3_^2−^ ion, where iron was in +4 oxidation state [37,38].

The process of partial disturbance of the structure of the interlayer gap as well as the main layers of the octahedral and tetrahedral network of the mineral matrix can occur during modification of the layered silicates by acrylamide-acrylic acid copolymer macromolecules. This can be reflected through broadening of the doublet lines and some increase in the magnitude of the quadrupole splitting of Fe^2+^ ions and through the transition of some Fe^3+^ ions to a more distorted position. The decrease in the chemical shift, which is equivalent to an increase in the density of s-electrons near the iron nucleus, and broadening of the splitting line were associated with a decrease in the symmetry of the iron ion environment. Thus, the amount of iron in the form of trivalent ions localized in octahedral (mainly in the M2 position) and tetrahedral positions was revealed to be more than half as much.

#### 3.1.3. XRD Analysis

Earlier [39], we presented the assumed structure of the biphasic hybrid composition. Figure 3 shows the X-ray diffractograms of bentonite (1), potassium hexaferrocyanide (PHF) (2), potassium hexaferrocyanide intercalated in the mineral matrix {BT-K_4_[Fe(CN)_6_]} (3) and copper ferrocyanide {BT-K_(4−x)_C_IIx_[Fe(CN)_6_]} (4) and percolated complexes P[AAm-AAc]:{BT} (5) and P[AAm-AAc]:{BT-K_(4−x)_Cu_x_[Fe(CN)_6]_} (6). The XRD data of bentonite (line 1) show characteristic diffraction reflections at 20.93°, 26.66°, 42.46° and 59.7°, which correspond to the diffraction reflections of quartz in the bentonite with interplanar distances (d) of 4.28 Å, 3.31 Å, 2.12 Å, and 1.54 Å, and also calcium, magnesium and aluminum silicates with frequent line overlap that makes their identification difficult.

This article explains some details related to the change in the structure of bentonite matrix, which includes fragments of quartz, montmorillonite, fluston, feldspar, halloysite, and mixed aluminum and magnesium silicates (Figure 3). This is expressed in the decrease or disappearance of diffraction band intensity for a number of noted silicates or aluminosilicates during the formation of percolated complexes of intercalated complex of BT-K_(4−x)_Cu_x_[Fe(CN)_6_] in the bulk of polymer matrix.

As indicated in the experimental section, the hybrid composition was prepared by an in situ polymerization method, in which bentonite and its intercalated complex with CuFC swollen in water were pre-impregnated with acrylic acid and acrylamide monomer solutions in the stirring mode. Acidic modification of bentonite can be assumed in this case. As a result, some aluminosilicate fragments were partially destroyed with the removal of alkaline and alkaline-earth cations from their crystalline lattice.

The analysis of diffractograms of the original bentonite and its percolated into P[AAm-AAc] (5) polymer matrix, ICC {BT-K_(4−x)_Cu_x_[Fe(CN)_6_]} (4) and P[AAm-AAc]:{BT-K_(4−x)_Cu_x_ [Fe(CN)_6_]} (6) indicated a drop in signal intensity at 2θ = 21.73 Å, 29.54 Å, 31.94 Å, 36.50 Å, 42.39 Å, and 52.88 Å, which on microphotography correlated with the decrease in the number of impurity flake clumps on the treated surface (Figure 4). This was also evidenced by the shift of the FT-IR absorption band from 1012 cm^−1^ to 1031 cm^−1^ and 1034 cm^−1^, typical of the valence vibrations of Si-O groups, and the reduced intensity of absorption bands at 760 cm^−1^ and 916 cm^−1^, characteristic of the valence vibrations of Si-O-Si and Al-O (OH)- groups in aluminosilicates, respectively.

The potassium hexacyanoferrate (PHC) had bands at 21.11° and 31.85° (line 3). For the intercalated complex of {BT:K_(4−x)_Cu_x_[Fe(CN)_6_]}, the characteristic diffraction bands appeared at 20.8° (d = 4.2803 Å), 26.63° (d = 3.3211 Å) and 31.85° (d = 2.8068 Å) (line 2). From a comparison of the diffraction bands of individual compounds (BT and PHC) and their complexes, it can be concluded that ICC should be considered as “embedded” copper hexaferrocyanide (CuHC) in the layers of bentonite without significant disturbance of the crystal lattice of the mineral matrix. A noticeable shift of the diffraction bands toward their reduction (from 21.11° and 26.72° to 20.81° and 26.63°, respectively) may indicate an increase in the interlayer distance between the aluminosilicate sheets of bentonite during the formation of a complex with hexaferrocyanides of mixed composition.

It should be taken into account that potassium, iron and copper ions, included in the mixed hexaferrocyanides, can be incorporated into the interplanar structure of layered silicate not only due to the weak adsorption interactions, since there are no steric structural geometric limitations, but also by means of chemical contacts of copper and iron in hexaferrocyanides with oxygen atoms of isomorphic silicon or aluminum oxides included in the bentonite structure. The lattice dimensions of bentonite and mixed copper ferrocyanide were estimated from the characteristic diffraction reflection bands and the values of interplanar distances. For bentonite d_20.8_^0^ = 4.2803 Å and ICC d_31.85_^0^ = 2.8068 Å, i.e., crystals K_(4−x)_C_x_[Fe(CN)_6_] as a part of the intercalated complex can be placed in the interlayer galleries of the layered mineral.

The retention of the diffraction band at 31.85 Å for both copper hexaferrocyanide and {BT:K_(4−x)_Cu_x_[Fe(CN)_6_]} can be explained by the fact that the mixed intercalated complex also contained individual crystals of potassium ferricyanide that did not interact with copper sulfate during formation of intercalated complexes. Such a conclusion was previously made in [19], according to which the data from chemical analysis evidenced that in the system CuSO_4_·K_4_[Fe(CN)_6_]·3H_2_O, a solid solution of K_4_[Fe(CN)_6_] in Cu_2_[Fe(CN)_6_] was formed continuously up to the composition 3Cu_2_[Fe(CN)_6_]·2K_4_[Fe(CN)_6_], which corresponds to the formula 4K_2_{Cu[Fe(CN)_6_]·Cu_2_[Fe(CN)_6_]}, and the presence of a compound of K_2_Cu[Fe(CN)_6_] composition was not confirmed. The conclusion that the coordination complex of K_2_Cu_3_[Fe(CN)_6_]_2_ composition is a solid solution between the compounds K_2_Cu[Fe(CN)_6_] and Cu_2_[Fe(CN)_6_] is doubtful. It is possible that the K_2_Cu_3_[Fe(CN)_6_]_2_ compound noted in [27,28] is isostructural with Cu_2_[Fe(CN)_6_], and it also can be interpreted as a solid solution of K_2_Cu[Fe(CN)_6_] with Cu_2_[Fe(CN)_6_].

As can be seen from Table 2, quartz is the least susceptible to acid modification, with preservation of the peak intensities at 2θ = 20.93° and 26.66° and the parameters of the crystal lattice, while other minerals undergo significant changes with the disappearance or displacement of characteristic bands on the X-ray patterns (Figure 3). The area of the silicon phase peaks in the bentonite diffraction pattern was calculated and compared with the JCPDS-ICDD database card (Table 3) at 2θ = 19.8°, 35.31°, 54.23° and 62.30°. It can be seen that the introduction of PHC and CuHC into the bentonite led to a noticeable decrease in the area under the marked diffraction lines.

This may indicate that the Fe^2+^ ions in the hexaferrocyanides take part in the formation of electrostatic bonds with the negatively charged aluminosilicate groups of the mineral matrix. However, at 2θ = 54.23° and 62.30°, there was a noticeable increase in the peak areas of the phase of diffraction patterns for the BT-K_(4−x)_Cu_x_[Fe(CN)_6_] system compared to BT-K_4_[Fe(CN)_6_]. It should be assumed that this may be defined by the fact that copper ions in the volume of CuHC can be located both in the nodes of crystal lattice and inside it.

The obtained data suggest the possible production of ICC and PCC with mixed structures consisting of crystalline (quartz, halloysite, aluminum and magnesium silicates) and non-crystalline phases, with their relatively uniform distribution over the volume of the polymer matrix.

Sequential modification of bentonite with copper ferrocyanide as well as the introduction of acrylamide and acrylic acid cross-linked copolymers into its volume by means of in situ polymerization promoted the formation of tortuous pores in the resulting hybrid material (Figure 4) and a significant increase in the number of heterogeneous and relatively small (from 500 nm to 2.5 microns) particles compared to the initial sample, where large globules over 5 μm in size could be observed. The observed texture corresponded to a loosened material of complex chemical composition with a simultaneous combination of natural quartz grains, crystals (bentonite, montmorillonite) and individual flake-like amorphous phases. The narrower size distribution for the hybrid composite is explained by the fact that bentonite micropowder acts as a dispersant in the inverse suspension polymerization process. The surface shapes of particles in Figure 4c,d are regular and rough compared with the particle in Figure 4a,b, which can result from irregular form and roughness of bentonite powders.

The resulting hybrid compositions had a well-developed porous surface, which contributed to the accelerated penetration into their volume of hydrated ions in the composition of LRW (Figure 5).

Thus, we synthesized a hybrid composite that included all of the most commonly and widely used sorption materials of natural (bentonite) and synthetic origin (hexaferrocyanides of transition metal) incorporated into a polymer matrix.

#### 3.1.4. Swelling Behavior

It is well known that composites based on cross-linked acrylic acid and acrylamide copolymers with clay particles are pH responsive, which is expressed as a significant increase in hydrogel swelling in water [40,41,42,43,44]. This behavior is defined by osmotic pressure exerted by the counter ions of negatively charged aluminosilicate layers of BT incorporated in the polymer matrix of P[AAm-AAc] and ionization of carboxylic groups of copolymers, as well as depending on pH media.

The hybrid compositions P[AAm-AAc]:{BT} and P[AAm-AAc]:{BT-K_(4−x)_Cu_x_[Fe(CN)_6_]} exhibited similar swelling behavior. As revealed by Figure 6, there was a gradual sweep in swelling up to a certain point, and then it became constant within 24 h. The swelling behavior of the hybrid composites was studied at various pH values in the range from 2.0 to 12.5 at room temperature. Analysis of Figure 6 data indicated that water absorption of both hybrid composites was sensitive to environmental pH.

Figure 6 demonstrates that the potentiometric curves of both hybrid composites clearly showed a conformational transition of linear chains of P[AAm-AAc] copolymer between crosslinking points in the range of pH between 4.2 and 7.5. At low pH, the swelling coefficient decreased as the carboxylate group on the polymeric matrix was protonated; then, the linear polymer chain between cross-linking points shrank and became hydrophobic. Additionally, the hydrogen bonds formed by unionized carboxylic and acrylamide groups along the polymer chains resulted in smaller pores. This, in turn, reduced the degree of ionization and, therefore, reduced the swelling coefficient. At a pH greater than the pK_α_ of acrylic acid (pK_α_ = 4.20), the repulsion between the negatively charged carboxylate ions resulted in larger pores. In an alkaline medium, the decrease in physical forces between acrylic acid and acrylamide groups additionally leads to a slightly smaller pore size. These results are consistent with the data of [40,45]. At high pH > 9, the swelling capacity is also reduced due to the “charge screening effect” of excess Na^+^ ions in the swelling medium, which protects the carboxylate anions and prevents effective anion–anion repulsions. In the pH range from 4.2 to 7.5 for P[AAm-AAc]:{BT} and from 4.2 to 9.5 for P[AAm-AAc]:{BT-K_(4−x)_Cu_x_[Fe(CN)_6_]}, some of carboxylic acid groups were ionized, and the electrostatic repulsion between the COO^−^-groups caused an enhancement of the swelling capacity. This behavior is typical for the “globule-coil” conformational transition of a hydrophobic polymer chain such as polymethacrylic acid or polyacrylic acid–surfactant complexes [46,47].

Thus, the incorporation of clay particles into the polymeric matrix increases the water absorbing capacity of the hybrid composite and allows them to be considered as superabsorbents with a high absorption capacity for metal ions, especially radionuclides.

It should be noted that the swelling capacity of P[AAm-AAc]:{BT-K_(4−x)_Cu_x_[Fe(CN)_6_]} hybrid composite was significantly higher than that of the percolated P[AAm-AAc]:{BT} one. Most likely, this was due to an increase in the osmotic pressure due to the excess charge created by mobile potassium ions in the intercalated complex BT-K_(4−x)_Cu_x_[Fe(CN)_6_] and mobile metal ions of bentonite, such as sodium, calcium, magnesium and aluminum.

The scanning electron microscope observations revealed that the composites had a heterogeneous layered structure (Figure 4c,d and Figure 5). A detailed and thorough inspection of the SEM images proved that the prepared composites had highly rough surfaces composed of deep meso- and macro-pores. These pores of different diameters clearly demonstrated the porous structure of the composite surfaces, reduced the internal diffusion resistance of the adsorbate ions, and thus increased the adsorption kinetics as well as the removal potential [40].

#### 3.1.5. Sorption of RN

As noted in the introduction, bentonite, transition metal hexaferrocyanides and their intercalated complexes are widely used for fixation and storage of active long-lived radionuclides from LRW and natural water bodies contaminated as a result of the activities of various branches of the nuclear energy sector [48,49,50,51,52].

In our previous article [39], we studied two-phase hybrid composite materials for immobilization and fixation of radionuclides in liquid radioactive waste (LRW) of the research water-water reactor WWR-K. It has been established that hybrid compositions have a highly synergistic effect on the sorption of ^137^Cs^+^ and ^134^Cs^+^ radionuclides. The distribution coefficient of cesium radionuclides in the hybrid composite materials is two times higher in comparison with the sorption activity in the mineral matrix.

Byproducts in LRW such as Co^2+^, Sr^2+^ and Cs^+^ are considered to be the most hazardous radionuclides to human health due to their high transferability, high solubility, long half-lives and easy assimilation in living organisms. Several technologies, such as chemical precipitation, conventional coagulation, reverse osmosis, ion exchange and adsorption, have been developed for removal of these radionuclides from wastewater [52,53,54]. Among these methods, sorption is the most widely used for removal of these radionuclides since it is simple and cost-effective with low-cost sorbents.

The sorption capacity is also related to the surface coverage of the adsorbent, which increases with the growth of adsorbate amount per unit volume of the solution at a constant mass of the adsorbent. The relationship between the number of metal ions adsorbed by a unit mass of adsorbent and the concentration of the remaining metal ions in the solution is represented by a sorption isotherm. In addition, the sorption isotherm also describes the sorption equilibrium, which reflects the degree of interaction between metal ions and the number of adsorption centers on the adsorbent.

At the first stage, model liquids of stable isotopes of metal ions, predominantly present in liquid radioactive waste, were studied to investigate the sorption of radionuclides by hybrid composites. The concentration of metal chloride in the model solution was 10^−4^ M, that is, two or three orders of magnitude higher than the concentration of radionuclides in natural LRW generated by the research water-water reactor BN-350. This study was performed as a model system for further discussion of the results of radionuclide sorption from real liquid radioactive waste generated in a research reactor.

The metal ion uptake efficiency as a function of contact time is shown in Figure 7. The metal ion sorption efficiency increased with contact time and reached equilibration after 2.5 and 3.0 days for the composite of P[AAm-AAc]:{BT-K_(4−x)_Cu_x_[Fe(CN)_6_]}. The metal ion sorption on the hybrid composite was very rapid, and quantitative sorption was observed within 24 h of contact time. In addition, the sorption efficiency was initially fast and reached 70% to 85% of the maximum capacity within 1.5–2 days. The results also showed that fast kinetics of Cs^+^ ions were observed on the hybrid composite compared to bi-(Sr^2+^, Co^2+^) and trivalent (Y^3+^) metal ions under the same experimental conditions. Figure 7 also shows that the efficiency of sorption of metal ions gradually decreased with time until saturation was reached. The slow process could be associated with the intraparticle diffusion of metal ions into the interior pores of P[AAm-AAc]:{BT-K_(4−x)_Cu_x_[Fe(CN)_6_]} composite. Hence, a contact time of more than 3 days was adopted in the following experiments for hybrid composites.

It is well known that under optimal conditions, the percentage of metal ion sorption depends on the ratio of the amount of adsorbate to the available active sites for ion exchange or complexation of the adsorbent [54,55].

Two-stage sorption was previously reported by other researchers [56]. The fast stage predominates quantitatively and, probably, due to the large number of active centers on the surface of the composites. The second stage is slow and quantitatively insignificant, consisting of gradual population of the active centers of the surface. Fast sorption can be associated with adsorption on the outer surface, which differs from the macro-pore adsorption process.

Preliminary experiments to determine the effect of the pH medium on the degree of binding of radionuclides showed that the most optimal pH value was an alkaline medium. As noted above, in an acidic medium, the negative charge is suppressed on the surface of both the mineral sorbent and the hybrid composition, which reduces the sorption of positively charged radionuclide ions both on the surface and in the internal cavities of the sorbents. As can be seen from Table 4, the hybrid composition had the highest sorption activity compared to the individual natural mineral, and its percolated form was in the body of the polymer matrix.

At pH > 8, interlayer cations in bentonite did not affect the sorption of radionuclides [57,58]. As a result, clay minerals of the 2:1 type with various interlayer cations penetrating into the polymer matrix, especially in the case of its intercalated complex with copper hexaferrocyanide, showed a sharp absorption of ^134^Cs^+^ and ^137^Cs^+^ radionuclides. It can be assumed that at pH > 9, ^60^Co^2+^ and ^57^Co^2+^ will precipitate as Co(OH)_2_. At the same time, it was reported [59] that the sorption of Co^2+^ on montmorillonite (the closest analogue of bentonite) increased significantly in the pH range from 5 to 8, which is consistent with our results, since the pH of the external solution decreased from 9 to 6 during sorption. The same feature was also observed in the case of RN sorption on the hybrid composite, since in this case, the carboxyl groups of the polymer matrix also took part in the complexation process according to schemes (A).

As revealed from Table 4, the distribution coefficient of RN, especially for cesium to the unmodified mineral sorbent, did not exceed ~30.0 mL·g^−1^ and was two orders of magnitude lower than that of the hybrid composite containing copper hexaferrocyanide. These data are in good agreement with the literature data [3,60,61]. The distribution coefficient of radionuclides for percolated sorbents decreased in the series ^134^Cs^+^, ^137^Cs^+^, ^60^Co^2+^, ^57^Co^2+^, and ^85^Sr^2+^. In the same order, a change in the sorption capacity was observed.

Thus, it is possible to present the following mechanism for binding of radionuclide ions with hybrid composition components: (A) complex formation with the -COOH groups of the polymer and ≡SiOH and ≡AlOH sites of mineral matrices with the release of protons, which explains the decrease in the pH of the medium; (B) ion-exchange process between sodium, calcium and magnesium ions of a natural mineral as well as cesium and strontium ions in external solution, respectively; (C) coordination binding of transition metal radionuclide ions (Co^2+^) with electron-donating oxygen and nitrogen atoms of the carboxyl and amide groups of the polymer matrix, respectively, and (D) incorporation into the cubic structure of copper hexaferrocyanide particles intercalated into clay, especially cesium ions, which is associated with the correspondence of the diameter of dehydrated Cs^+^ with the size of the entrance windows of transition metal hexaferrocyanides and the minimum solubility of cesium and strontium in hexaferrocyanide complexes. All noted types of radionuclide interactions with the components of the hybrid composite are presented in Figure 8.

The resulting sorption data show that sorption of metal ions by each radionuclide was suppressed by others. The maximum values of sorption capacity in LRW solution were higher for ^134^Cs^+^ and ^137^Cs^+^ as well as ^60^Co^2+^ and ^57^Co^2+^, while the sorption of ^85^Sr^2+^ was suppressed. All sorption processes are spontaneous in nature, according to the negative values of Gibbs energy [58]. 

The positive value of enthalpy (ΔH) for Sr^2+^ indicated an endothermic sorption process; however, sorption of Co^2+^ and Cs^+^ was exothermic, i.e., showed ΔH values. The sorption of Co^2+^ onto sorbents was strongly pH-dependent, and surface complexation was considered as a dominant sorption mechanism. We assume that the sorption of Sr^2+^ was suppressed by competition with Co^2+^ since the complexation ability of a transition metal ion is higher than that of alkali-earth iron due to the additional donor–acceptor bond between the electron-donating functional groups of polymeric matrix (-COOH and -NH_2_C=O) and d-electron orbitals of Co^2+^.

## 4. Conclusions

The synthesis and characterization of new biphasic hybrid composite materials consisting of intercalated complexes (ICC) of the natural mineral bentonite with copper hexaferrocyanide (phase I) that were incorporated into the bulk of the polymer matrix (phase II) and their sorption abilities toward radionuclides of liquid radioactive waste (LRW) were investigated.

Four mechanisms of radionuclide metal ion binding with the components of hybrid composition were presented.

The information obtained during this study makes it possible to obtain new classes of hybrid composites for effective sorption of certain types of radionuclides at the interface between mineral and polymeric matrices. This will allow the use of hybrid materials as highly effective sorption materials for radionuclide fixation of LRW.

This work was performed in the framework of Grant under the Scientific-Technical Program (State Registration No. 0121RK00777) funded by the Ministry of Education and Science of the Republic of Kazakhstan for 2021–2023 on the topic “Applied research based on a nuclear reactor in the area of radioactive waste management, radioisotopes production and radiation materials science”.

## Figures and Tables

**Figure 1 polymers-15-02586-f001:**
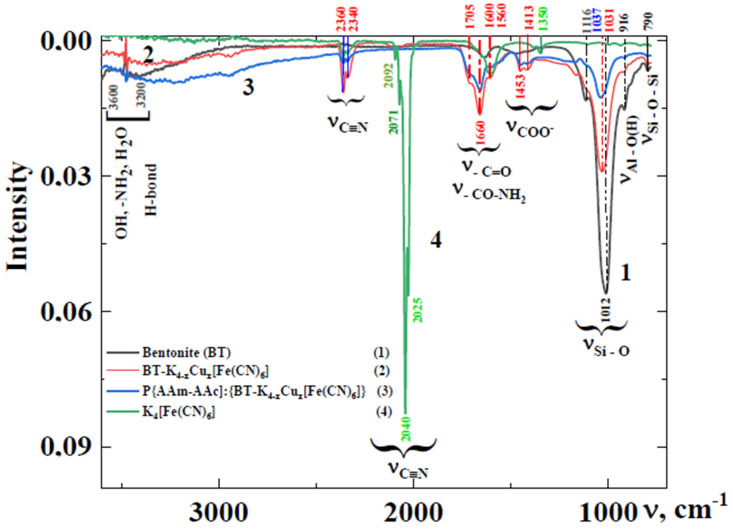
FT-IR spectra of BT (1), BT-K_(4−x)_Cu_x_[Fe(CN)_6_] (2), P[AAm-AAc]:{BT-K_(4−x)_Cu_x_[Fe(CN)_6_]} (3) and K_4_[Fe(CN)_6_] (4).

**Figure 2 polymers-15-02586-f002:**
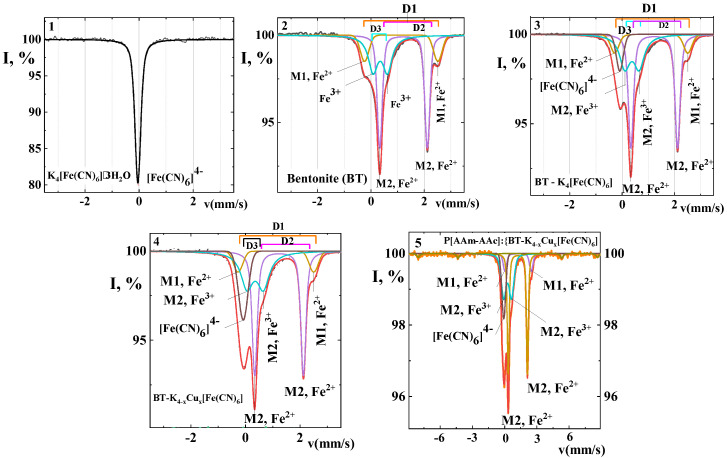
Mössbauer spectra of K_4_[Fe(CN)_6_] (1), BT (2), BT-K_4_ [Fe(CN)_6_] (3), ICC BT-K_(4−x)_Cu_x_[Fe(CN)_6_] (4), PCC P[AAm-AAc]:{BT-K_(4−x)_Cu_x_[Fe(CN)_6_]} (5).

**Figure 3 polymers-15-02586-f003:**
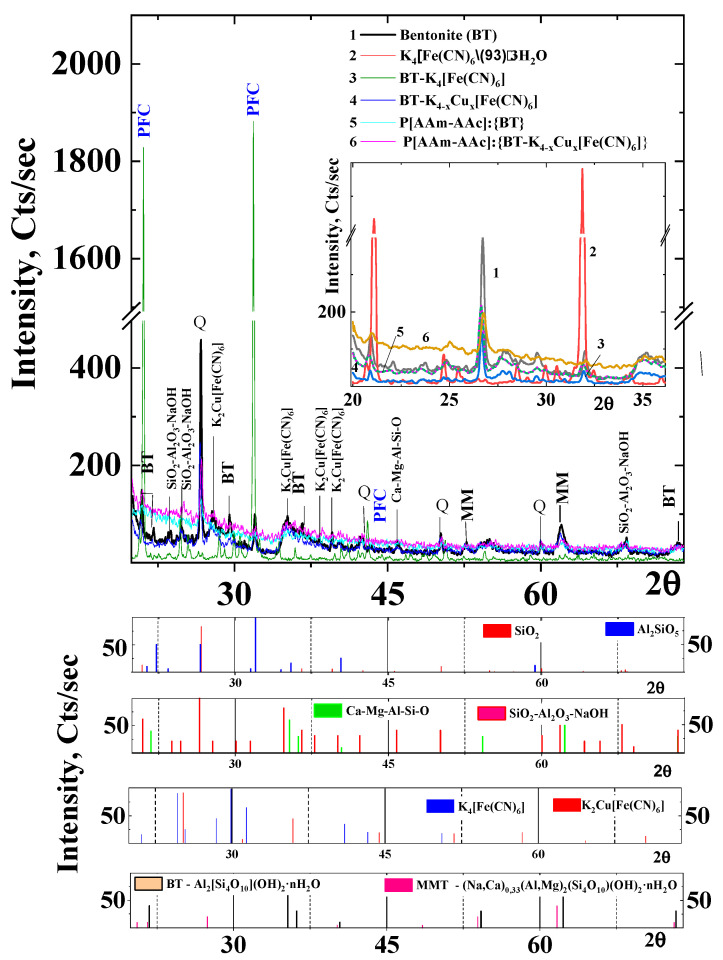
XRD patterns of BT (1), K_4_[Fe(CN)_6_] (2), ICC BT-K_4_ [Fe(CN)_6_] (3), BT-K_(4−x)_Cu_x_[Fe(CN)_6_] (4), PCC P[AAm-AAc]:{BT} (5) and P[AAm-AAc]:{BT-K_(4−x)_Cu_x_[Fe(CN)_6_]} (6).

**Figure 4 polymers-15-02586-f004:**
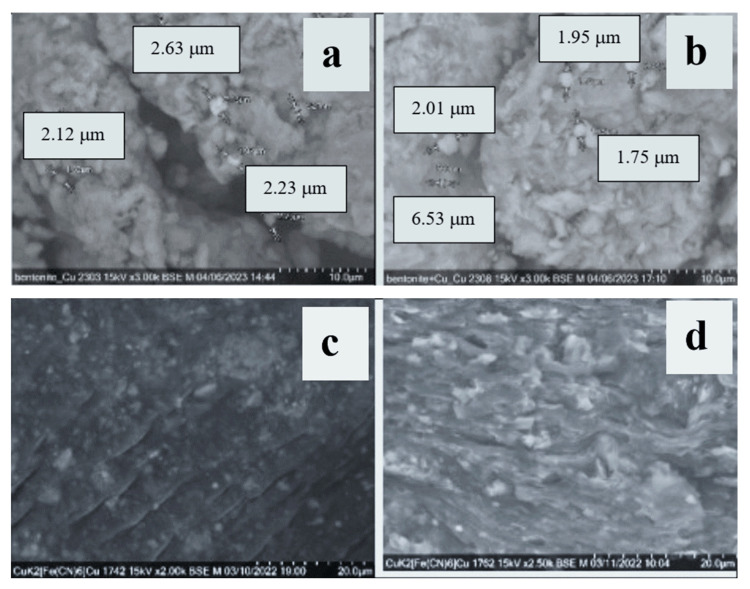
SEM micrograph of BT (**a**), ICC BT-K_(4−x)_Cu_x_[Fe(CN)_6_] (**b**), PCC P[AAm-AAc]:{BT} (**c**) and P[AAm-AAc]:{BT-K_(4−x)_Cu_x_[Fe(CN)_6_]} (**d**).

**Figure 5 polymers-15-02586-f005:**
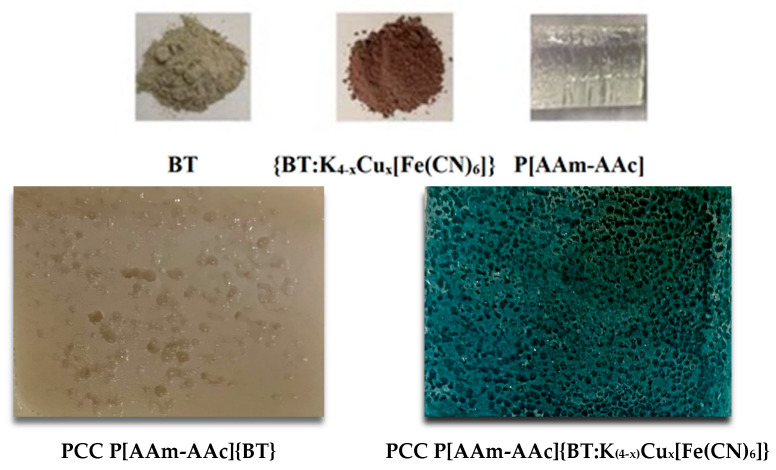
Images of initial components BT, K_4_[Fe(CN)_6_], ICC BT:K_4-x_Cu_x_[Fe(CN)_6_], PCC P[AAm-AAc]{BT} and P[AAm-AAc]{BT:K_(4−x)_Cu_x_[Fe(CN)_6_]}.

**Figure 6 polymers-15-02586-f006:**
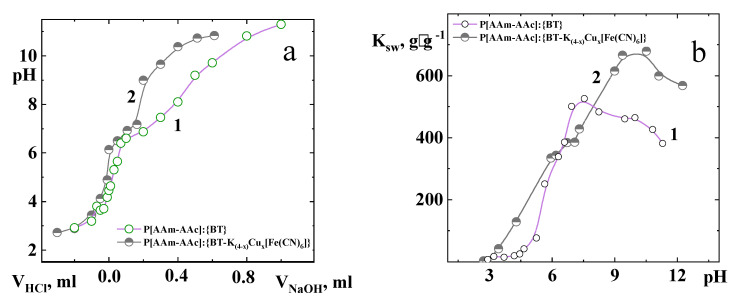
Potentiometric titration data (**a**) and swelling coefficient (**b**) dependence of P[AAm-AAc]:{BT} (1) and P[AAm-AAc]:{BT-K_(4−x)_Cu_x_[Fe(CN)_6_]} (2) on pH media.

**Figure 7 polymers-15-02586-f007:**
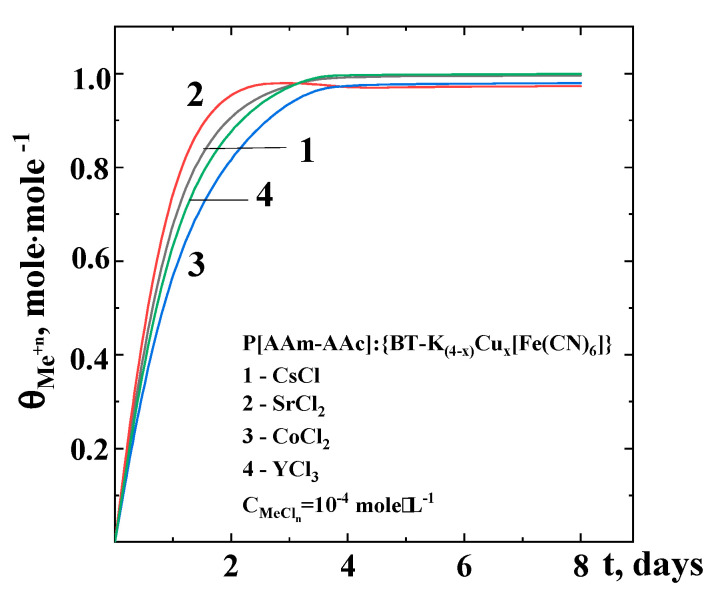
Sorption isotherm of analogues of radionuclide metals onto P[AAm-AAc]:{BT-K_(4−x)_Cu_x_[Fe(CN)_6_]} composite.

**Figure 8 polymers-15-02586-f008:**
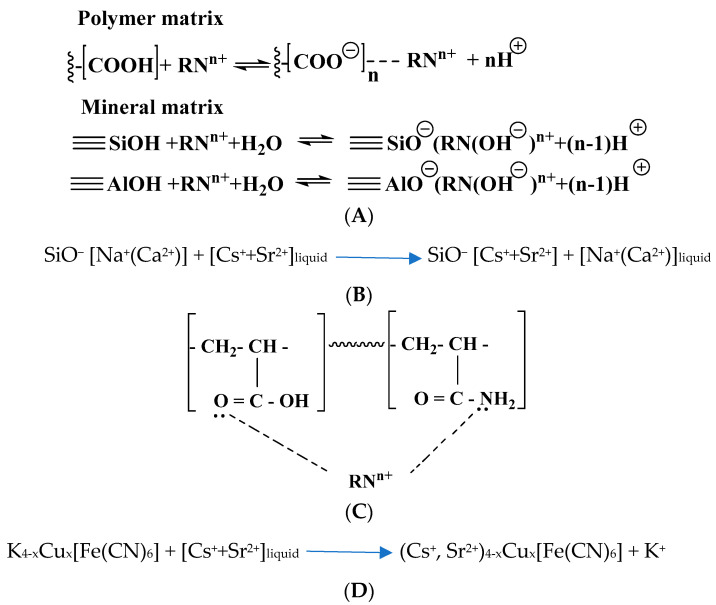
Scheme of radionuclide sorption by P[AAm-AAc]:{BT-K_(4−x)_Cu_x_[Fe(CN)_6_]} composite. (**A**) complexation mechanism; (**B**) ion exchange mechanism; (**C**) donor–acceptor mechanism; (**D**) incorporation mechanism.

**Table 1 polymers-15-02586-t001:** Mössbauer spectroscopy data for K_4_[Fe(CN)_6_], BT, ICC BT-K_(4−x)_Cu_x_[Fe(CN)_6_] and PCC P[AAm-AAc]:{ BT-K_(4−x)_Cu_x_[Fe(CN)_6_].

Subspectrum	I, %	δ, mm/s	Δ, mm/s	Assignment
K_4_[Fe(CN)_6_]·3H_2_O				
S	100	−0.045	0	[Fe(CN)_6_]^4−^
Bentonite				
D1	50.3	1.225	1.785	Fe^2+^, M1
D2	15.7	1.137	2.750	Fe^2+^, M2
D3	34.0	0.351	0.538	Fe^3+^, M2
BT-K_4_[Fe(CN)_6_]				
D1	48.7	1.227	1.787	Fe^2+^, M1
D2	10.8	1.163	2.694	Fe^2+^, M2
D3	34.2	0.311	0.656	Fe^3+^, M2
D4	6.30	−0.045	0	[Fe(CN)_6_]^4−^
BT-K_(4−x)_Cu_x_[Fe(CN)_6_]				
D1	42.8	1.231	1.7866	Fe^2+^, M1
D2	9.6	1.117	2.772	Fe^2+^, M2
D3	31.1	0.354	0.602	Fe^3+^, M2
D4	16.4	−0.082	0	[Fe(CN)_6_]^4−^
P[AAm-AAc]:BT-K_(4−x)_Cu_x_[Fe(CN)_6_]				
D1	9.3	1.252	2.444	Fe^2+^, M1
D2	37.6	1.230	1.787	Fe^2+^, M2
D3	32.6	0.277	0.714	Fe^3+^, M2
D4	15.7	−0.079	0	[Fe(CN)_6_]^4−^
D5	4.90	0.395	−0.186	FeO_3_^2−^

**Table 2 polymers-15-02586-t002:** XRD data for quartz in BT, ICC BT-PFC and BT-CuFC, from the JCPDS-ICDD database card No. 85-0795 SiO_2_.

Card No. 85-0795	BT	BT-K_4[_Fe(CN)_6_]	BT-K_(4−x)_Cu_2_[Fe(CN)_6_]
a = 4.10, Å	4.917	4.912	4.910
C = 5.402, Å	5.398	5.406	5.408

**Table 3 polymers-15-02586-t003:** Areas of reflections of bentonite, from the JCPDS-ICDD database card No. 03-0019 Bentonite (Na-Al-Si-O-OH-H_2_O).

File Name	d = 4.46903 A, θ = 19.801°	d = 2.56022 A, θ = 35.307°	d = 1.69848 A, θ = 54.231°	d = 1.49764 A, θ = 62.302°
Bentonite pure	46.98	66.73	21.04	73.58
B-K_4_[Fe(CN)_6_]	38.19	57.43	11.80	47.29
B-Cu_2_ [(Fe(CN)_6_]	33.96	49.39	19.42	54.57

**Table 4 polymers-15-02586-t004:** Radionuclide sorption characteristics of bentonite, [AAm-AAc]:{BT} and P{AAm-AAc]:{BT-K_(4−x)_Cu_x_[Fe(CN)_6_]} composite (sorption time is 8 days).

	Degree of RN Removal, %					K_d_, mL·g^−1^				
	^134^Cs^+^, Bq·L^−1^	^137^Cs^+^, Bq·L^−1^	^60^Co^2+^, Bq·L^−1^	^57^Co^2+^, Bq·L^−1^	^85^Sr^2+^, Bq·L^−1^	^134^Cs^+^, Bq·L^−1^	^137^Cs^+^, Bq·L^−1^	^60^Co^2+^, Bq·L^−1^	^57^Co^2+^, Bq·L^−1^	^85^Sr^2+^, Bq·L^−1^
BT	59.4	51.2	38.4	34.5	28.7	26.8	23.4	37.2	36.4	14.6
P[AAm-AAc]:{BT}	63.9	55.8	88.4	86.6	34.8	61.36	44.07	26.15	22.1	16.9
P[AAm-AAc]:{BT-K_(4−x)_Cu_x_[Fe(CN)_6_]}	99.7	99.3	90.4	88.8	21.2	3118.9	1258.7	131.9	118.0	5.36

## Data Availability

Ref. [39]. The work is in progress.

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
