# Peer review of "Synthesis and Investigation of the Properties of Biphasic Hybrid Composites Based on Bentonite, Copper Hexacyanoferrate, Acrylamide and Acrylic Acid Hydrogel"

_polymers, 2023, doi:10.3390/polym15122586_

Round 1

Reviewer 1 Report

This paper by Mamytbekov et al describes the utilization of hybrid composite materials like intercalated complexes (ICC) for the utilization of toxic radioactive chemicals. The synthesis process and the overall presentation including advanced characterizations are quite okay and impressed me a lot. But, I found some minor issues in the manuscript which requires rectification. If authors are willing to revise and make a point-wise answer of all my comments and questions, then I would be pleased to consider this as revision. Please find my comments below:

1. If the work is based on adsorption, there should be some focus on the porous characteristics of the composite, which is missing in this study. Authors are encouraged to add on such BET results or so on which describes the adsorption of radioactive elements.

2. K4[Fe(CN)6] shows a marginal and almost abolished band of CN in the composite polymer materials, why ?  

3. This needs further justifications: in page 12, “This may indicate that the Fe2+ ions in the hexaferrocyanides may take part in the formation of electrostatic bonds with the negatively charged alumosilicate groups of the mineral matrix.”

4. How the potentiometric titration data were collected, authors need some explanations in it.

5. Are the layer structure of the composite confirmed from their XRD analysis ?

6. Page 16 has a formatting error. Please check.

7. In Page 16, Please explain, “The metal ions sorption on hybride composite was very rapid and quantitative sorption was observed within 30 min contact time.”

8. If  the electromotive force may be responsible for the sorption of radioactive sites, then what is the necessity to make such a complex composite ? Why not a simple system is required? Please explain.

9. In the introduction section, the significance of these materials as well as other realistic composites and polymers could draw the wide attention of the readers. Authors should consider this while writing the introduction section. They should also cite relevant papers on adsorbents of toxic metals and radioactive elements. Langmuir 2023, 39, 11, 4071–4081.

NA

Author Response

Dear Reviewer,

Thank you very much for the work you have done, which has improved the presentation and quality of the submission.

Reviewer 2 Report

The manuscript entitled with “Synthesis and Investigation of the Properties of Biphasic Hybrid Compositions Based on Bentonite, Copper Hexacyanoferrate, Acrylamide and Acrylic Acid Hydrogel” by Mamytbekov et, al. reported a biphasic material consisting of inorganic complexes and polyacrylamide/polyacrylic acid for its sorption abilities for radionuclides in liquid radioactive waste. The authors also investigated the sorption mechanism of this material. In addition, the authors provided detailed discussion, but that might be too detailed/too wordy and should be revised for easy reading. Overall, I would recommend its consideration to publish in Polymers if the writing and some technique issues have been addressed and improved.

1. The title of the manuscript is extremely long and confusing. The authors should shorten it down and revise it carefully.

2. The abstract must be improved. Its current form is too long with too many paragraphs while its structure is off topic.

3. The authors need to pay attention to their writing as long sentences with clear grammar mistakes are used, in addition to poor structure is applied. The authors must also improve this section and get clear logic trend/origin for their work. Language is somehow important for readers to follow the authors and to better understand the work. 

4. I do not quite agree with the authors that their materials are biphasic hybrid compositions. First of all, the inorganic complex applied onto the polymeric matrix may not necessarily to be a different phase from the polymer network. 

5. Figure 2 is of very poor quality. It must be improved. Also, I have no idea what are the colored curves for in Fig 2-2 (2-3, 2-4 & 2-5).

6. What about the initial concentration of RNs in the LRW? And how much maximum amount of LRW can be processed per weight of the material?

7. Can the hybrid material be reused? To make the material useful, both the points in 6 and 7 should be considered.

Need to be improved

Author Response

(The authors gave the same response as above.)

Reviewer 3 Report

as attached

as attached

Author Response

(The authors gave the same response as above.)
